# Treatment of Mine Water with Reverse Osmosis and Concentrate Processing to Recover Copper and Deposit Calcium Carbonate

**DOI:** 10.3390/membranes13020153

**Published:** 2023-01-25

**Authors:** Alexei Pervov, Htet Zaw Aung, Dmitry Spitsov

**Affiliations:** Moscow State University of Civil Engineering, 26, Yaroslaskoye Highway, 129337 Moscow, Russia

**Keywords:** reverse osmosis, nanofiltration, sludge dewatering, sludge moisture, calcium carbonate deposition on seed crystals

## Abstract

Mine water usually contains heavy metals and other inorganic and organic pollutants that contaminate water bodies. Reverse osmosis (RO) techniques are capable of producing purified water that meets discharge regulations. However, the problem of RO concentrate disposal and utilization is still not solved. The well-known zero liquid discharge (ZLD) process provides total concentrate utilization at the power industries but seems unreasonably expensive for the treatment of large amounts of mine water due to required chemical softening and the evaporation of concentrate. In the present article, a new approach to increase the recovery of reverse osmosis and to avoid high operational costs is demonstrated and discussed. The new technique involves radical RO concentrate flow reduction and withdrawal, together with dewatered sludge. The idea to “hide” concentrate in dewatered sludge is proposed and demonstrated during experiments. The article demonstrates results of the conducted experimental program aimed at reduction of volumes of all liquid wastes produced during mine water treatment using a new approach to concentrate it with a cascade of nanofiltration membranes and to reach a TDS value of 110–120 g per liter. The obtained concentrate is mixed with the wet sludge, which is further dewatered and withdrawn together with the dewatered sludge. Experiments are conducted that demonstrate a reduction in calcium in the concentrate due to deposition of calcium carbonate on the “seed crystals” in the circulation mode. Another distinguishing feature of the new technique is the separation of concentrate into two streams containing high concentrations of monovalent ions (sodium and ammonium chlorides) and divalent ions (calcium, magnesium and copper sulphates). Flow diagrams of the processes are presented to demonstrate the water treatment technique used to produce deionized water and two types of sludges: sludge after clarification and sludge after calcium carbonate deposition.

## 1. Introduction

### 1.1. Treatment of Mine Water with Reverse Osmosis, Elimination of Concentrate, and the ZLD Process

Mining provides an enormous amount of pollution due to discharges of saline water with heavy metals. These discharges are rich with sulphates, hardness, ammonia, oil and heavy metals, and have high TOC concentration values. The main efficient tool to handle these water discharges is to use RO to purify the water and to produce clean water that meets discharge regulations, which is then added to other water bodies. Yet, the problem of concentrates arise when we apply RO for water treatment. To eliminate concentrates of RO membrane facilities used in industrial water treatment, a ZLD process is introduced that involves chemical softening of RO concentrate and it’s further evaporation. However, high energy costs and the large volumes of water to be treated make this solution very expensive. In addition, the situation with water mining treatment is also complicated by the fact that mining industries are located far in the north, and reagent supply also requires extremely high costs. In this article, the authors have undertaken the task of developing a new technique to achieve optimization and reduction in operational costs by reaching high recoveries in RO facilities. This solution involves the experience collected by the authors throughout extensive research undertaken during the last decade to understand scaling and scale inhibition mechanisms in RO. The main concept of the technique consists of the production of a small amount of concentrate, i.e., a reasonable amount to evaporate, to produce dry salty sediment that can be further forwarded to a special landfill [1]. The salty sediments contain sparingly soluble salts, heavy metals (aluminum and copper), ammonia, sodium chloride and sodium sulphate. Analysis of the mining water chemical composition (Table 1) prompted us to realize that such a complicated approach as the ZLD process can be avoided, and the technique can be simplified.

-The regulation requirements for calcium, magnesium, sulphate and TDS content in discharged water is not very strict, and necessity to apply a high rejection RO process is doubtful.-The copper content in the mine water is low; nonetheless, the concentration of this metal is three–four times greater than the regulation value.-There are a number of techniques developed by authors of this article that enable us to reduce RO concentrate flows to the values equal to those of water volumes contained in dewatered sludges after water clarification and softening processes.-It is unlikely that application of the ZLD process can reach high water recoveries and avoid high operational costs to supply chemicals for water softening; therefore, the authors suggest using the newly developed technique to remove hardness from RO concentrate using NF techniques operated in circulation mode [1,2]. The authors suggest a novel technique to treat mining water and utilize concentrate without the use of softening chemicals and evaporation. The new technique involves a dramatic reduction in concentrate flow value to withdraw concentrate and all impurities rejected by membranes, together with dewatered sludge, which is forwarded to the landfill. “In a nutshell”, below, we provide the main considerations that formed the basis of the technology developed and presented in this article.

Concentrate disposal and possible utilization still remains an unsolved problem for many cases when concentrate discharge is not available. There are also cases where existing developed techniques to reduce and eliminate concentrate are extremely expensive, and the project cannot afford such approaches due to their high costs and the limited availability of power and chemicals. The company that have ordered this project had only one solution based on the use of RO for mine water purification and the use of the ZLD process to dispose of concentrate discharge. Assuming that the location of the project is in the north, high energy costs and unavailability of reagent delivery have led to customers looking for new solutions to the problem.

To solve the problem of concentrate discharge and to escape additional costs, as well as the use of softening and evaporation (that are required when the ZLD method is used), the authors have implemented four new techniques. This approach consists of reducing concentrate flow by 100–200 times and hiding this volume in the dewatered sludge, thus withdrawing all rejected impurities, together with dewatered sludge, and treating it as sludge moisture. To reach extremely high recoveries and avoid calcium carbonate deposition in membrane channels, a new approach is developed that involves the constant deposition of calcium carbonate on “seed crystals” without constant addition of softening reagents (such as lime). To efficiently utilize concentrate, another new approach is used that enables us to separate concentrate flow into two concentrated flows: a flow primarily containing monovalent ions and flow containing heavy metals.

The treatment of mine water remains the unsolved problem. It is partly solved by the application of RO, as reverse osmosis membranes efficiently reject heavy metals and other pollutants dissolved in water. However, treatment by RO has the complication of concentrate disposal. A number of projects have used the published ZLD techniques that require the evaporation of concentrate. To efficiently evaporate concentrate with minimal power costs, full softening of the concentrate is required (removal of calcium), which utilizes lime consumption and pellet reactors. For many customers located in the far north, such reagent deliveries, as well as power consumption, are very expensive.

In our project, the company that serves Norilsk Nickel addressed us to develop an alternative solution to treat RO concentrate without high expenses due to reagent deliveries and evaporation. Initially, the project involved the ZLD process: reagent softening of RO concentrate and further evaporation to obtain wet salts. To solve the problem, the authors used four techniques developed and described previously [2,3,4], that enable us to reduce and “hide” RO concentrate.

-Reduction in concentrate volume to a value that does not exceed 0.4–0.5% of the initial feed water volume and withdrawal of concentrate together with the dewatered sludge;-Reduction in concentrate volume and deposition of calcium carbonate using “seed crystals”;-Reduction in concentrate volume using a cascade of low-rejection membranes;-Separation of concentrate into two solutions, i.e., monovalent salt and divalent salt concentrated solutions, to facilitate its further utilization.

### 1.2. Removal of Calcium Carbonate from RO Concentrate

The main problem associated with handling and reducing RO is calcium carbonate scaling. A method was previously developed that dramatically reduces concentrate volumes [2]. The concentrate is softened due to mixing with calcium carbonate seed crystals in a reactor, while the RO unit is operated in recirculation mode with constant product flow withdrawal. This process is described in detail in [2]. Using this approach for feed water composition, given in Table 1, and applying RO treatment, we can reduce concentrate volume by 100–150 times as compared to the feed water flow value. Moreover, using nanofiltration (NF) membranes with low rejection characteristics, we achieve high recovery rates and high TDS of concentrate [3,4]. As shown in [2,3,4], the use of the developed process enabled us to remove up to 75–80% of dissolved calcium from the feed water.

Figure 1 demonstrates a pilot membrane unit containing three stages of brine concentration and deposition of calcium carbonate in the seed reactor [2].

### 1.3. A New Approach to “Hide” Concentrate in Dewatered Sludge

The second distinguishing factor of the newly developed technique is the idea of “hiding” concentrate in the dewatered sludge. This technique was successfully implemented when RO was used for the treatment of fugate after sludge dewatering using a centrifuge and the recycling of removed impurities by returning them to the sludge prior to its dewatering [3]. Usually, the sludge, after water clarification and water softening, is dewatered using filter press equipment or centrifuges. Dewatering provides a value of sludge moisture up to 80%. Depending on the feed water composition, sludge amount after dewatering is about 0.1–0.15%, and the proportion of water in this sludge is 0.3–0.45%. To maintain the salt balance (or other dissolved matter balance), the amount of salt removed by the RO system and discharged together with RO concentrate flow should equal to amount of salt withdrawn with the dewatered sludge. Thus, we should achieve a concentrate flow value of 0.3–0.4% of the feed water flow. In other words, concentrate flow should be 1/300 to 1/250 of the feed water flow. In our case, we have two separated processes and two types of sludge: sludge after clarification and sludge after calcium carbonate removal. Assuming we have equal amounts of sludge and 80% moisture in both cases, we can expect that the volume of concentrate will be 0.7–0.8% of feed water flow, thus we reduce the initial feed water volume by 120–130 times using membrane facility producing demineralized water and concentrate, the TDS value of which reaches 170–180 g per liter [2,3,4]. Photos of laboratory test membrane units are shown in Figure 1 and Figure 2.

The idea of “hiding” RO concentrate in dewatered sludge and in calcium carbonate slurry is based on the material balance of salts (or other impurities contained in water) that enter the dewatering facility (thickening tank) and salts withdrawn from the water treatment plant together with dewatered sludge. All dissolved salts are supplied with RO concentrate; thus, the concentrate flow rate should not exceed the flow rate of water withdrawn with the sludge as the sludge moisture. Thus, the concentrate flow rate should correspond to water flow rate withdrawn with the sludge, and all dissolved salts should be withdrawn with this water amount. The balance flow diagram of the technological scheme of mine water treatment is shown on the Figure 3. Figure 4 shows this balance in an example when feed water flow equals to 1000 cubic meters per hour. Usually, a typical water clarification plant produces 5 cubic meters of dewatered sludge per 1000 cubic meters of treated feed water. The dewatered sludge moisture value is 80%, which means that, in dewatered sludge, 4 cubic meters of water per 1 ton of solids are present. Thus, the concentrate volume should correspond to 0.4% of the feed water volume. In other words, the feed water volume should be reduced (or concentrated) 250 times by RO. For our case, where feed water TDS is 1000 ppm, we can reach 120–150 g per liter in concentrate, using a pressure value of 16 bars [1,2], or reduce feed water flow by 120–130 times. Thus, 0.4% of concentrate (with 120,000 ppm TDS) is withdrawn together with the dewatered sludge and other 0.4% are withdrawn together with dewatered slurry (calcium carbonate sludge).

### 1.4. Principles to Reduce Concentrate Flow

The concentrate flow is treated using low-rejection NF membranes. To reach high TDS values, a “cascade” of NF membrane stages is used to reach the desired concentration without high energy costs [4] using pressure values that do not exceed 20 bars (Figure 5). Principles of brine concentration using low pressures and low-rejection membranes are discussed in detail in [4].

Figure 2 shows three-stage membrane units developed for brine concentration to reduce concentrate flow rate to a value not exceeding 0.5–1% of the feed water entering the unit. The developed by the authors, RO units producing 4 cubic meter per hour (a) and 20 cubic meter per hour (b) are presented in Figure 2.

### 1.5. Separation of Heavy Metals

The last problem is that of heavy metals [4,5]. The presence of heavy metals in sludge changes the sludge hazard category and makes their storage in landfill dangerous [6,7,8]. There are a number of different approaches developed to separate copper from other salts [9,10,11]. It seems reasonable to add concentrated copper solution to the sludge after water clarification that contains aluminum, and to then forward this sludge to a special landfill. The second half of the dewatered sludge (that contains calcium carbonate) can be left free of copper and utilized separately as an industrial raw material. It is possible to separate RO concentrate into the different components depending on their rejection characteristics [4,5]. Studies have already been published [12,13,14,15,16,17] describing how to separate concentrate into organic solution and inorganic salt concentrated brine [1,4]. In our case, the concentrate contains copper. As feed water volume was reduced by 100–150 times throughout RO treatment, the copper concentration substantially increased to the concentration of 4–5 ppm, which exceeds regulation limits for storage at municipal landfills. There are well-known equations that describe how to calculate concentration values of different components in permeate and concentrate [4].

The following equation calculates the concentration in permeate:(1)$$C_{p}=C_{f}\times \left(1-R\right){\left(1-\alpha \right)}^{-R}$$
where R is rejection of the ion; α is recovery; C_f_ is concentration in the feed water; and C_p_ is concentration in permeate.

The concentration of this component in concentrate is illustrated by Equation (2):(2)$$C_{c}=C_{f}\times {\left(1-\alpha \right)}^{-R}$$

Using these equations, we can evaluate the ratio of concentrations of different ions with different valencies both in permeate and concentrate depending on their rejection qualities. In our case, we used nanofiltration membranes of the 70 NE model (produced by CSM); rejection of monovalent ions (sodium, chloride and ammonia) was 70% and rejection of multivalent ions (such as calcium, sulphate and copper) reached 90%. Thus, if we take recovery to be 0.9 (i.e., if we reduce feed water volume by 10 times), we can calculate their ratio value in concentrate as
(3)$$\frac{C_{{NH}_{4}^{+}}^{c}}{C_{{Cu}^{2+}}^{c}}=\frac{C_{{NH}_{4}^{+}}^{f}}{C_{{Cu}^{2+}}^{f}}\times {\left(1-\alpha \right)}^{R_{Cu}-R_{Na}}$$
and in permeate, the ratio value will be the following:

Assuming the ratio value C_Cu_/C_NH_4__ in the feed water as 1, we see that in concentrate, this value increases by 2 times, and in permeate it decreases two times respectively. To further change this ratio and increase the content of copper in the concentrate, we can dilute this concentrate 10 times using distilled water or RO permeate, then treat this solution using the same test membrane unit (Figure 6), and again, reach a recovery value of 0.9. The reached values of concentration ratios in concentrate and permeate are described by Equations (4) and (5), respectively:(4)$$\frac{C_{{NH}_{4}^{+}}^{p}}{C_{{Cu}^{2+}}^{p}}=\frac{C_{{NH}_{4}^{+}}^{f}}{C_{{Cu}^{2+}}^{f}}\times {\frac{1-R_{NH_{4}}}{1-R_{Cu}}(1-\alpha )}^{R_{Cu}-R_{NH_{4}}}$$
(5)$$\frac{C_{{NH}_{4}^{+}}^{c}}{C_{{Cu}^{2+}}^{c}}={\left(1-0.9\right)}^{0.2}=0.63$$
(6)$$\frac{C_{{NH}_{4}^{+}}^{p}}{C_{{Cu}^{2+}}^{p}}=\frac{1-0.65}{1-0.85}\times 0.1^{0.2}=2.33\times 0.65=1.47$$
(7)$$\frac{C_{{NH}_{4}^{+}}^{C^{2}}}{C_{{Cu}^{2+}}^{C^{2}}}={\left(1-0.9\right)}^{0.2}\times 0.63=0.63^{2}=0.4$$

We can then apply another dilution cycle and obtain the following results:(8)$$\frac{C_{{NH}_{4}^{+}}^{p^{2}}}{C_{{Cu}^{2+}}^{p^{2}}}=\frac{1-0.65}{1-0.85}\cdot {\left(1-09\right)}^{0.2}\times 2.33-0.63=2.33^{2}-0.63^{2}=2.17$$

We can further increase the copper content in concentrate by applying the third dilution cycle, and reach
(9)$$\frac{C_{{NH}_{4}^{+}}^{c^{3}}}{C_{{Cu}^{2+}}^{c^{3}}}={\left(1-0.9\right)}^{0.2}\times 0.63\times 0.63=0.63^{3}=0.25$$
(10)$$\frac{C_{{NH}_{4}^{+}}^{p^{3}}}{C_{{Cu}^{2+}}^{p^{3}}}=\frac{1-0.65}{1-0.85}\times {\left(1-0.9\right)}^{0.2}\times 2.33^{2}\times 0.63^{2}=2.33^{3}\times 0.63^{3}=3.15$$

Thus, by applying dilution cycles and further concentrating the water, we can increase the concentration of multivalent ions in concentrate and decrease the concentration of monovalent ions. In permeate, we obtain the inverse concentration ratio values. Concentration values of separated components can be further increased by applying the concentration process using RO and NF membranes.

Finally, due to the development of nanofiltration membranes, it has now become possible to concentrate discharged brines and effluents for the purpose of their further utilization [18,19,20,21,22,23,24,25,26]. In particular, there is a lot of information on the use of nanofiltration for the purification of centrifuges after the dewatering of mineralized sediments of natural and waste waters [27,28,29,30,31,32]. The authors of this article analyzed the studies already carried out at the Department (water supply and wastewater treatment system) of MGSU University [33,34] on the use of reverse osmosis and nanofiltration processes not only for the purification of wastewater (silt water) for the dehydration of sewage sludge and the purification of filtrates from solid waste landfills [21,33,34], but also for the disposal of membrane plant concentrates and removal of all contaminants trapped by the membranes together with the dehydrated sludge [31,32,33]. The new development (Figure 3 and Figure 4) involves the use of a membrane plant for the simultaneous purification of the wastewater and the utilization of the concentrate of the reverse osmosis plant with guaranteed production of high-quality purified water. The concentrate from the membrane plant is subsequently sent to the sludge thickener tank.

### 1.6. Principles of Flow and Material Balance of Concentrate Utilization

Principles of the balance calculation of flows and concentrations shown in the scheme of Figure 7 are based on the observance of the material balance. During the wet sludge dewatering process using a centrifuge, about 2% of the sludge is rejected, and only 0.4% of this amount is withdrawn with the dewatered sludge as the sludge moisture. The rejected substance (also called fugate) is usually discharged into the sewer. In our example, considering that feed water flow rate is 1000 cubic meters per hour, respectively, the centrifuge consumption will be 20 cubic meters per hour, and the amount of water removed with dewatered sludge per hour will equal 4 cubic meters. In order to withdraw all rejected impurities with the dewatered sludge, it is necessary to maintain a material balance: the amount of salt contained in concentrate that enters the dewatering system should be equal to the amount of salt withdrawn together with the dewatered sludge. Thus, with feed water salinity of 1500 ppm and expected concentration value for product water of 500 ppm, the total amount of salt in the concentrate will be 1000 kg. For the concentrate flow value of 4 cubic meters per hour, salt concentration in the concentrate will be 250 g per liter, and for a concentrate flow rate value of 8 cubic meters per hour, the concentrate TDS will be 125 g per liter. In our project, in order to reach total concentrate utilization, we applied a new technology of calcium carbonate deposition on seed mass. Thus, part of the concentrate will be removed together with dewatered calcium carbonate sludge. We have chosen a total concentrate flow value of 8 cubic meters per hour with a concentrate TDS “reasonable” value of 120 g per liter, which is easy to reach using 16 bars pressure and nanofiltration membranes [2,33,34]. With 4 cubic meters of water removed with the sludge after feed water clarification, the salinity value of this water will be 120 g per liter, and the salinity of concentrate flow will also have a value of 120 g per liter. In addition, the value of the total salt content of water entering centrifuge 3 after thickener 1 should also be 120 g per liter (Figure 4). Thickener 1 receives two streams: 4 cubic meters per hour of concentrate and 24 cubic meters per hour of sludge, with a total salt content of 1500 mg/L. Therefore, the reverse osmosis unit operated in the circulation mode must treat the mixture of these streams (4 plus 24) to produce 24 cubic meters per hour of with TDS value of 200 ppm and to reach the TDS value of concentrate 120 g per liter at it’s flow rate of 4 cubic meter per hour.

Main goals of this article were

To produce water to meet regulation standards to discharge in surface water sources.To develop measures to utilize concentrate.To develop tools to withdraw concentrate together with dewatered sludge after clarification, and together with sludge after softening.To develop techniques that require minimum chemicals and operational costs.To use developed techniques to reduce concentrate flow; remove sparingly soluble salts without chemicals using recirculation mode; withdraw all impurities rejected by membranes together with dewatered sludge; to separate heavy metals from monovalent ions.

## 2. Experimental Section: Materials and Methods

The purpose of the experiments was to study membrane rejection and flux characteristics, and to demonstrate the possibility of reducing concentrate volume by 100–120 times as well as to separate copper and monovalent ions to produce two separate concentrate streams. The experimental program includes three series:Evaluation of rejection characteristics and obtaining of dependencies of different ion concentration values in product water and concentrate on K (coefficient of initial volume reduction) value s. Evaluation of required K value to meet required regulation values in the purified water.To reach a K value of 100 (reduce feed water volume by 100 times) throughout the test run to demonstrate calcium carbonate deposition without the use of softening chemicals.To separate produced concentrate volume of 1 liter into two volumes that contain concentrated copper and ammonia.

In total, 200 L of the mine water effluent from a mine called “Limestones” in Norilsk region (Russia) was delivered to laboratory. The water composition is presented in the Table 1.

The flow diagram of the test unit is shown in Figure 7. The feed water was placed in tank 1 and then was delivered by pump 2 to membrane module 3. In the membrane module, the feed water stream was separated into two streams: permeate (purified water) and concentrate. The concentrate was returned back to tank 1 and the permeate was forwarded to tank 4. The Procon rotary pump provided a supply of 180–200 L per hour with a pressure value of 16 bars. The experiments were carried out using membrane elements of the 1812 standard model supplied by Toray Advanced Material Korea Inc. (the manufacturer of CSM membrane technologies), with reverse osmosis membranes (BLN model, 95–96% salt rejection) and nanofiltration membranes (70 NE model, 70% salt rejection). The area of the membranes in the 1812 membrane was 0.5 square meters. As the permeate was accumulated in tank 4 during test run, the volume of feed water in tank 1 decreased, while the concentrations of contaminants—dissolved salts and organic substances, generally estimated by the COD value—increased, and the product flow of membrane module decreased. The experimental procedure enables us to determine the efficiency of different species rejection, to evaluate sparingly soluble salts scaling rates on membrane surfaces, as well as to evaluate organic fouling of membranes, and to select membrane type for use in industrial application. Through the course of the experimental program, the dependencies of concentrations of various contaminants on the coefficient K value were determined. The initial volume reduction coefficient K is determined as the ratio of the volume of feed water to the volume of concentrate in the tank 1 (Figure 7) at the end of the experiment. The value of K corresponds to the membrane unit recovery indicator, which is equal to the ratio of the permeate flow rate to the flow rate of the feed water Q_p_/Q_f_, through the ratio Q_p_/Q_f_ = 1 − 1/K.

COD values were determined using the titrimentic method. Dissolved Organic species concentrations were determined using the method of spectrophotometry. An atomic adsorption spectrophotometer of the “A2” model was used (supplied by “Carl Stuart Group”, Lenster, Ireland, UK). Calcium and magnesium concentrations were determined trilometrically. Determination of sulphate ion concentrations were carries out using the turbidimetric method. Concentration values of ammonia ion were determined using the photometric method. A photoelectric photometer of the “KFK-3-01-Z” model was used (produced by “ZOMZ”, Sergiev-Posad, Russian Federation). Sodium ion concentrations were determined using the atomic adsorption method and the “dry residue” was determined using the weight method. Electric conductivity, TDS and temperature values were determined using laboratory conductivity meter, model Cond.730 (WNW “Inolab-Akvilon”, Moscow, Russian Federation). pH values were determined using laboratory pH meter HI 2215 (Hanna Instruments, Vohringen, Germany).

At the first stage of the experiments, the operation of the membrane module on the first stage of the membrane scheme was simulated (Figure 5), and the selection of membrane type for the first stage was made. Three experiments were performed as a first step. The feed water volume in each experiment was 10 L, and three experiments were conducted: using nanofiltration membranes without the addition of antiscalant to the feed water; using nanofiltration membranes with the addition of antiscalant to the feed water; using reverse osmosis membranes with the addition of antiscalant to the feed water. Throughout the first series of experiments, the initial feed water volume was concentrated 10 times and permeate volume collected in tank 4 (Figure 7) reached 9 L. Concentrations of calcium, chloride, sulphate, bicarbonate, copper and ammonia, as well as TDS and COD values, were tested throughout test runs. The further processing of the collected data enabled us to evaluate the rejection of different species by the membranes, and to select membrane type and recovery on the first stage to reach the required product water quality that meets regulation standards for discharge into water bodies. Furthermore, calculation of calcium concentrations throughout the test runs provided an evaluation of scaling rates in membrane modules on the first stage. The second step of the first experimental series included concentrating of the feed water 10 times using the nanofiltration membrane, further treatment of the produced permeate with reverse osmosis and nanofiltration membranes to produce water that meets regulation standards, and comparison of the efficiencies of the two membrane types. The feed water volume in the second series was 160 L. Antiscalant “Aminat-K” was added to the feed water with a dose of 5 ppm. The initial feed water volume of 160 L was reduced throughout the test run to the value of 16 L. Permeate (140 L) was collected in the permeate tank 4 (Figure 7) and used at the third step of the experimental series to test the second stage membrane treatment efficiencies. Dependencies of different species concentration values on K values in nanofiltration membrane concentrate and permeate are presented in Figure 8a,b. In the third step, the performance of nanofiltration and reverse osmosis membranes at the second stage of the membrane treatment scheme was evaluated and compared. The first stage nanofiltration permeate was treated by nanofiltration and reverse osmosis membranes in the circulation mode. The feed water (NF permeate) volume in both experiments was 20 L. The feed water volume was reduced by 10 times throughout both test runs. Results of various species evaluations in concentrates and permeates depending on K values are presented in Figure 9, Figure 10 and Figure 11. The results of the calcium carbonate scaling rate evaluation for RO and NF membranes during the first stage of the membrane treatment scheme are shown in Figure 12.

The second step of the experiments was aimed to demonstrate the possibility of reducing the initial feed water volume by 100–120 times, and to withdraw calcium carbonate from concentrate using its growth on the seed crystal surface [2]. This technique was developed by the authors and tested in a number of projects [2]. This new method was proposed as a solution to problems in mining water treatment and concentrate reduction, as it is a suitable technique to avoid water softening chemicals consumption [2]. The feed water volume in the second series of experiments was 16 L, which is the volume of concentrate left after the first series of experiments. Antiscalant “Aminat-K” was added to the feed water at a dose of 5 ppm. During the first series conductance, feed water volume was reduced by 10 times to simulate the first stage of the membrane treatment scheme (Figure 3, Figure 4 and Figure 5). To simulate calcium carbonate growth on the seed mass, the seed crystals were produced by the deposition of calcium carbonate after the addition of caustic solution to concentrate. In brief, 220 mL of caustic (NaOH) 1 N solution was added to 8 L of concentrate (half of the concentrate volume). Calcium carbonate was then deposited. The settled water after calcium carbonate deposition was removed, and crystals were added to the concentrate. To simulate the process of calcium carbonate deposition in the seed reactor, the deposited calcium carbonate was added to the second half of the first stage concentrate. Addition of the seed mass to concentrate causes a reduction in calcium concentration [2], which is reflected by the drop of calcium concentration in Figure 13. After calcium was reduced, the crystals were sedimented and filtered by filter paper. Both halves of the concentrate were mixed, and concentrating process was continued. After concentrate volume was decreased by 2 times (reduced from 16 to 8 L), contact with the seed crystals was repeated by passing concentrate through the crystal bed. After calcium was partly removed, concentrate volume was again reduced by two times from 8 to 4 L, and then, after contact with seed crystals, reduced from 4 to 2 L. Results of calcium ion concentration dependencies on K values after contacting the seed crystal bed are shown in Figure 13. Figure 14 shows the reduction in nanofiltration membrane-specific product flow throughout feed water concentration.

The third series of the experimental program was conducted to demonstrate the possibility of separating heavy metal multivalent ions and monovalent ions to facilitate further utilization of the concentrate. The presence of heavy metals in the concentrate (in our case, the presence of copper) in the sludge of suspended matter and in calcium carbonate sludge creates problems with sludge utilization as it increases the value of the hazard category. To reduce the amount of the sludge forwarded to the hazardous landfill, we attempted to separate the concentrate into two concentrated solutions: a solution with copper and a solution without copper. The solution with copper can be withdrawn together with sedimented matter sludge, and the monovalent ion concentrate is utilized together with the calcium carbonate slurry. In the third series, we diluted the concentrate produced in the second series by 10 times. Reverse osmosis permeate was added to 2 L of concentrate to increase the volume of the solution to 20 L. Then, the solution was treated using nanofiltration membranes (Figure 7). At the first step, concentrate volume was reduced to 2 L, and 18 L of permeate was collected in tank 4 (Figure 7). In the second step, concentrate in tank 1 was diluted again by deionized water, the volume increased to 20 L, and concentration cycle was repeated. At the third step, again, deionized water was added to tank 1 to increase the concentrate volume to 20 L, and again, the solution was concentrated by 10 times. Results of copper and ammonia concentrations in concentrate throughout the third series are shown in Figure 15.

## 3. Discussion of the Experimental Results

Figure 8a shows the concentrations of chlorides, sodium and sulphate ions in nanofiltration module permeate in terms of their dependency on K. Concentrations of copper and ammonia, as well as COD values, as functions of K are shown in Figure 8b. The first stage permeate is treated in the second stage either by reverse osmosis (a) or by nanofiltration membranes (b), as shown in Figure 9. The recovery in the second stage, or K value in the second stage, is selected based on the required rejection value of certain impurities, which should not exceed the regulation value in the product water. In our case, the copper concentration in the mine water should be reduced by 7–8 times and the ammonia value should be reduced by 4 times. Figure 13 and Figure 14 present concentrations of ionic species rejected by membranes as a ratio of concentration of the contaminant in the feed water to the regulation value of its concentration in water permitted to be discharged into the surface water body. If the ratio value exceeds 1, the purified water quality will not correspond to the regulation value. Figure 10 and Figure 11 show the dependencies of these specific concentration ratio values of ammonia and copper (Figure 13) and concentrations of other ions and TDS values (8) on K values in the reverse osmosis membrane permeate. The K value that corresponds to the intersection point in Figure 10 when the copper concentration value exceeds the regulation value was selected to determine the second stage recovery. Based on the selected K value, we can determine permeate TDS and concentrations of other species (Figure 11). Figure 12 shows the results of the calcium carbonate growth rate evaluation, which confirm that selection of the nanofiltration membranes in the first stage provides safe operation of the membrane facility. Figure 12 demonstrates the efficiency of the developed calcium reduction technique based on the use of seed crystals. The graphs in Figure 13 confirm that a substantial reduction in calcium concentration is reached without the constant consumption of softening chemicals as calcium carbonate deposition occurs, due to the constant maintenance of the supersaturation value through the operation of the membrane facility in the recirculation mode. The detailed description of the process is given by the authors in a number of publications [2]. Figure 14 shows the reduction in membrane specific product flow due to the growth of concentrate TDS.

The results of copper ion separation are presented in Figure 15. It can be seen that, from cycle to cycle, divalent copper remains in the concentrate due to the high rejection value (a), and monovalent ions leave the concentrate solution through the low rejection nanofiltration membrane (b). Thus, after all the “manipulations” with the concentrate, we obtain a concentrate with a high concentration of copper (about 70% of all copper contained in the feed water) and a permeate that contains 70% of monovalent salts. This solution can be further concentrated using RO and NF membrane cascades to reach a TDS value of 120 g per liter. The flow diagram of the concentrate separation process is presented in Figure 16. Concentrate collection, dilution and concentration is implemented in tank 1 using pump 2 and nanofiltration membrane 3 (first stage membrane) operated in recirculation mode. Permeate of membrane 3 is collected in tank 4 and is again treated by nanofiltration membrane 6 (second stage membrane), using pump 5 to concentrate copper and return it to tank 1 by pump 7. Membrane 6 product flow is collected in tank 8, pumped (9), and then concentrated by the cascade of reverse osmosis membrane 10 and nanofiltration membrane 11 to reach a high TDS value of 120,000 ppm. Permeate of reverse osmosis membrane 10 is used for concentrate dilution in tank 1, and permeate of nanofiltration membrane 11 is forwarded to tank 8 to mix with the feed water of the third stage. The first stage concentrate is treated by nanofiltration membrane 3 after the third dilution/concentration cycle until the TDS value of 120,000 ppm is obtained. This concentrate is forwarded to sludge thickening tank 1 (Figure 4) to be withdrawn together with dewatered sludge and aluminum. This high-hazard-category sludge is sent to a special landfill. Concentrate with low copper content is supplied to the sedimentation tank with calcium carbonate slurry 2 (Figure 4), and is then withdrawn, together with dewatered calcium carbonate sludge, to be used as raw materials in the industrial market.

Figure 5a,b show the steps to reach high TDS of concentrate and to reduce its volume. Figure 10 and Figure 11 show values of different ions and impurity concentration values that enable us to predict chemical composition. Calcium carbonate deposition on the seed crystals reduces TDS and facilitates further concentration increase. Experiments show that high TDS can be reached and concentrate volume can be reduced by a value of less than 1% of the initial volume. This amount can be shared between sludge dewatering facilities for suspended sludge dewatering and calcium carbonate precipitate. Figure 4 shows flow diagrams and mass (salt) balance during sludge dewatering to ensure the withdrawal of all excessive salts and rejected impurities together with dewatered aluminum sludge and precipitated calcium carbonate.

The proposed approach facilitates the treatment of mine water with a reverse osmosis plant, the utilization of concentrate by reducing its volume to a value less than 0.5–1% of initial feedwater volume, as well as the withdrawal of rejected salts and impurities with dewatered sludge and slurry.

The economic considerations of the project account for the savings acquired via this new technology of concentrate utilization. The approach does not provide liquid wastes as it separates mine water into clean purified water flow discharged into water bodies and dewatered sludge that is forwarded to landfill. The technological scheme also lacks chemical softening and evaporation of concentrate. Therefore, this approach saves operational costs.

The separation of concentrate and separation of aluminum sludge and slurry permits the use of sludges separately, reducing the cost of using special landfills to store high-hazard-category sludge.

Reduction in costs was the main goal of the research. According to the initial (conventional) approach, the mine water was clarified using coagulants and then was treated by RO to remove contaminants. The RO concentrate was then softened to remove calcium, and further concentrated by 10–20 times to reach an amount that does not exceed 3–5% of the initial feed water flow. Then, the concentrate was evaporated, with the end product of wet salts, which were forwarded to a special landfill together with dewatered sludge after the clarification stage. Such an approach involves substantial operational costs, including the purchase of reagents (about 300 g of lime per one cubic meter of the feed water, about 0.3 USD per 1 cubic meter) as well as additional power (about 22–25 kilowatts per one cubic meter of evaporated concentrate) that corresponds to additional use of 1.5–2 kilowatts per 1 cubic meter of the feed water, which corresponds to additional costs of 0.25 USD. Total savings per 1 cubic meter of the feed water treatment are 0.55 USD.

The new described approach proposed by the authors uses neither the lime softening process nor evaporation. The described deposition of calcium on the seed requires less than 2% of the stoichiometric amount [2].

## 4. Conclusions

A technological scheme is developed and described to treat mine water using membranes that facilitates the utilization of concentrate after membrane treatment by its concentration and withdrawal with dewatered sludge. The dewatered sludge after mine water pretreatment and clarification contains aluminum and copper, and is forwarded to a special landfill.Based on the results of experimental data processing, balance diagrams of the sludge dewatering process are developed that enable us to withdraw all rejected and concentrated impurities with dewatered sludge.Experimentally obtained relationships are presented that allow us to choose the optimal type of membrane at each stage and predict the composition of purified water at each membrane stage.A new technique is demonstrated to reduce the flow of reverse osmosis membrane concentrate with simultaneous removal of calcium that is deposited on the “seed crystals”. The technique enables us to reach high concentrate TDS and to remove up to 80% of calcium hardness without using excessive stoichiometric amounts of softening chemicals.A new approach is demonstrated to reduce the amount of utilized RO concentrate that contains hazardous heavy metals, with the example of copper removal. Concentrate solution is separated into two streams containing either divalent ions (including copper) or monovalent ions.

## Figures and Tables

**Figure 1 membranes-13-00153-f001:**
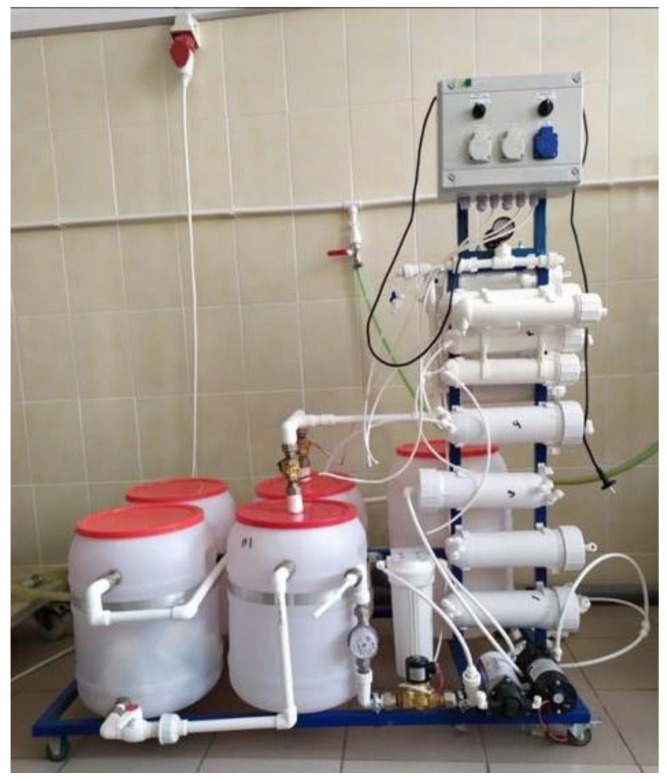
A three-stage pilot membrane unit for brine concentration using deposition of calcium carbonate in the seed reactor in continuous mode.

**Figure 2 membranes-13-00153-f002:**
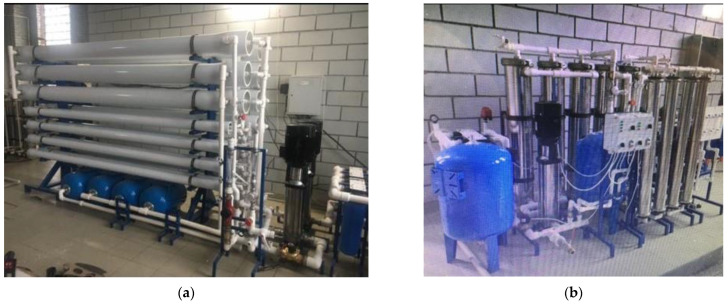
Reverse osmosis membrane units producing 4 cubic meters per hour (**a**) and 20 cubic meters per hour (**b**) of quality water with brine concentration membrane devices to reduce concentrate flow rate to 1% of the feed water flow rate or less.

**Figure 3 membranes-13-00153-f003:**
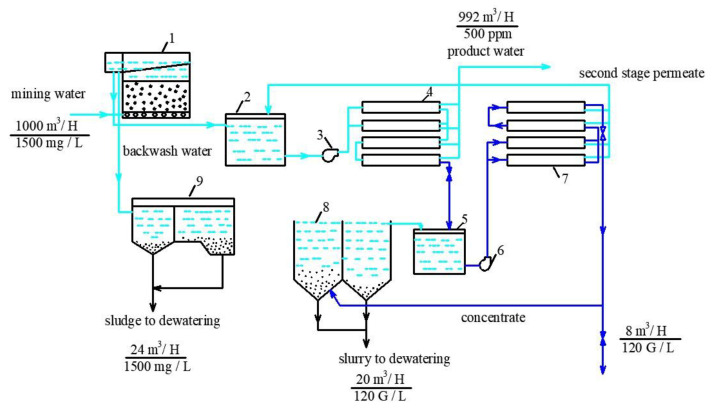
A flow diagram of mining water treatment, sedimented sludge withdrawal and concentrate utilization together with calcium carbonate slurry: 1—contact clarifier; 2—clarified water collection tank; 3—RO facility working pump; 4—RO membrane modules array; 5—first stage concentrate collection tank; 6—second RO stage working pump; 7—second NF membrane modules array; 8—seed crystals reactor; 9—sludge thickener.

**Figure 4 membranes-13-00153-f004:**
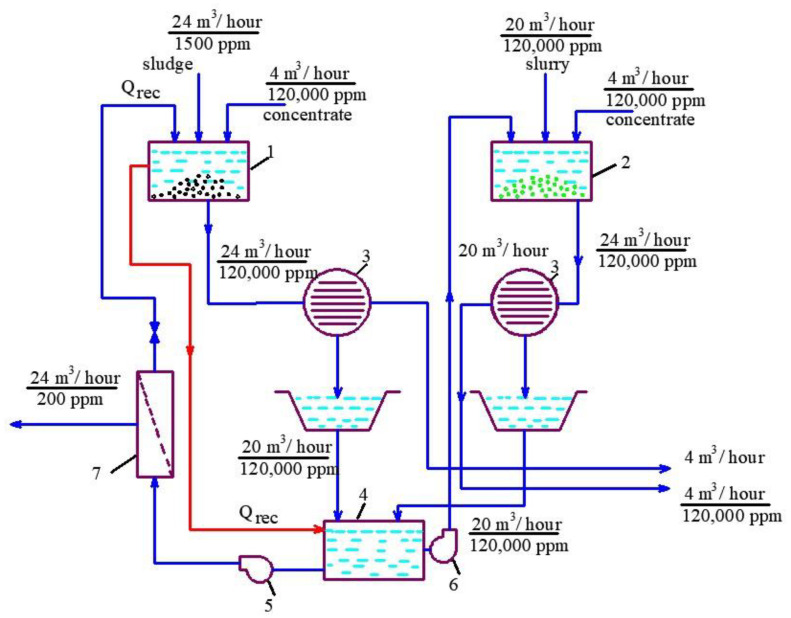
Balance schemes of dewatering of sludge after water clarification and calcium carbonate slurry after concentrate softening and treatment of reject effluents with RO to withdraw all removed impurities with the dewatered sludge: 1—sludge thickening tank; 2—calcium carbonate sedimentation tank; 3—centrifuge; 4—reject effluent collection tank; 5—working pump for RO plant; 6—pump to return reject back to sedimentation tank; 7—RO membrane modules.

**Figure 5 membranes-13-00153-f005:**
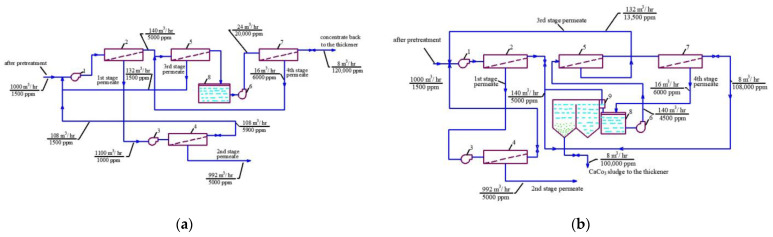
Salt balance flow diagrams of mining water treatment with RO and NF membranes and principles of reduction in concentrate flow by 100–200 times (**a**) without application of seed crystals and (**b**) with the use of a seed reactor: 1—working pump of membrane plant; 2, 5, 7—membranes, respectively, on the first stage of membrane facility, on the third and on the fourth steps of the feed water concentrating process; 3—the second stage of membrane facility working pump; 4—membrane on the second stage of membrane facility; 6—working pump of the second membrane stage; 8—fourth stage concentrate collection tank; 9—seed reactor.

**Figure 6 membranes-13-00153-f006:**
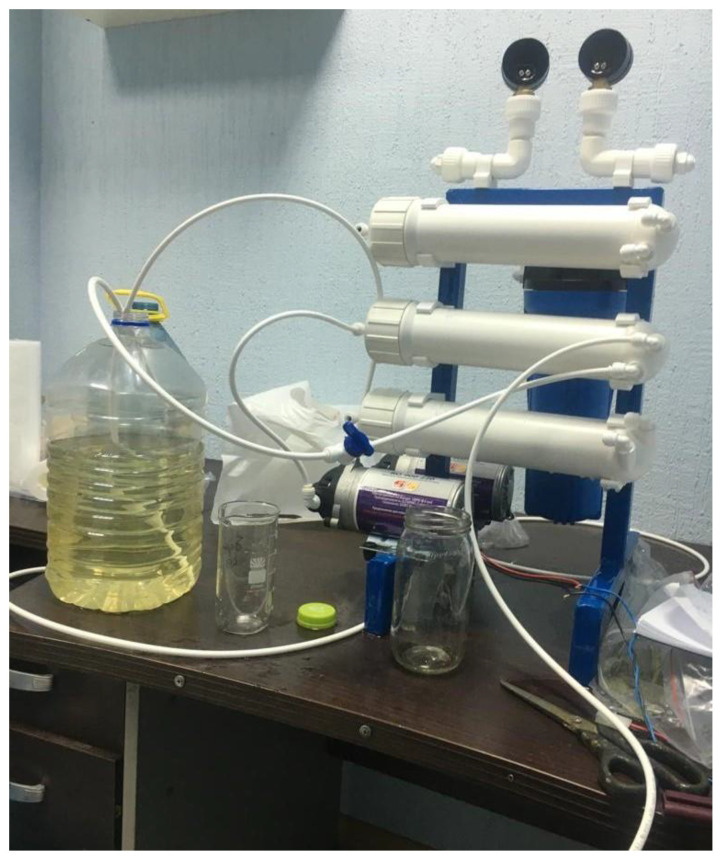
Membrane test unit with different membrane types to investigate separation of the brine into monovalent ion concentrated solution and divalent ion concentrated solution.

**Figure 7 membranes-13-00153-f007:**
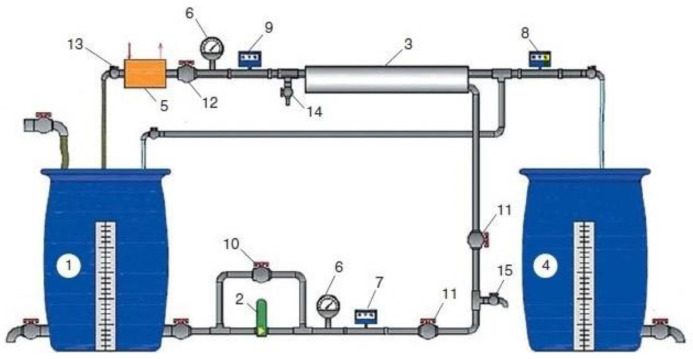
Experimental membrane unit flow diagram: 1—feed water tank; 2—pump; 3—membrane element in the pressure vessel; 4—permeate tank; 5—heat exchanger; 6—manometer; 7–9—flow meters; 10—bypass valve; 11—valve for adjusting the flow of source water; 12—valve for adjusting the working pressure and concentrate flow; 13—valve for adjusting the flow of cooling water; 14, 15—samplers.

**Figure 8 membranes-13-00153-f008:**
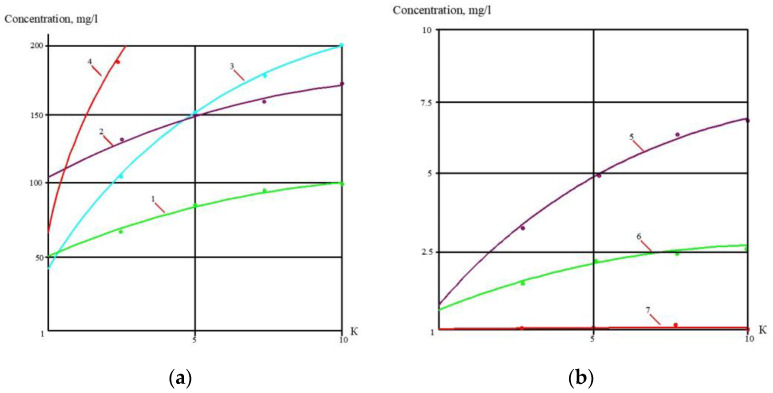
Dependencies of concentrations of aluminum, calcium, chlorides and sulphates, as well as COD value, on K values in concentrate (**a**) and pereate (**b**) of NF membranes in the first stage: 1—chloride; 2—sodium; 3—calcium; 4—sulphates; 5—COD; 6—ammonia; 7—copper.

**Figure 9 membranes-13-00153-f009:**
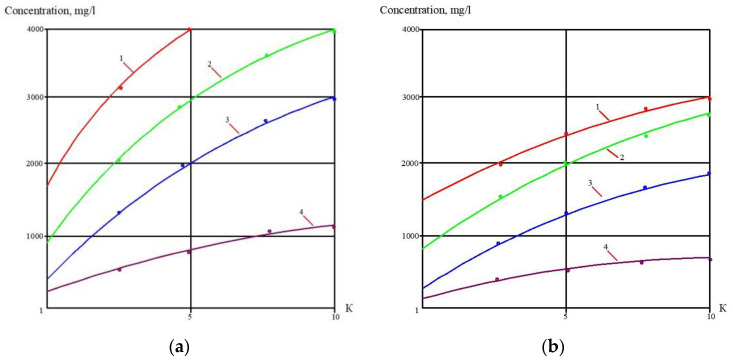
Dependencies of concentration values of chloride, sodium, calcium ions, as well as TDS value, on K in concentrate of reverse osmosis (**a**) and nanofiltration (**b**) membranes in the first stage of membrane treatment: 1—chlorides; 2—sulphates; 3—calcium; 4—chloride.

**Figure 10 membranes-13-00153-f010:**
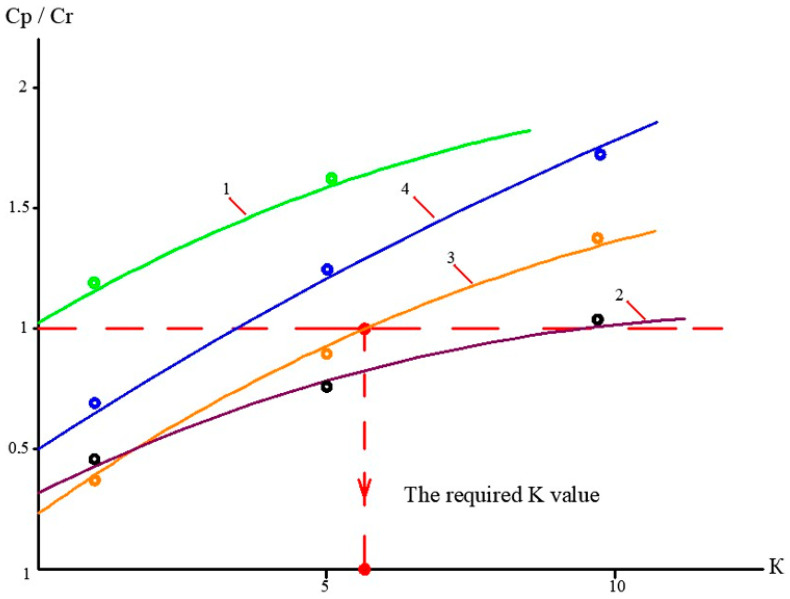
Dependencies of Cp/Cr ratio values on K values for different membranes used on the first stage of membrane treatment: 1—ammonium ions in RO permeate; 2—ammonium ions in NF permeate; 3—copper ions in RO permeate; 4—copper ions in NF permeate.

**Figure 11 membranes-13-00153-f011:**
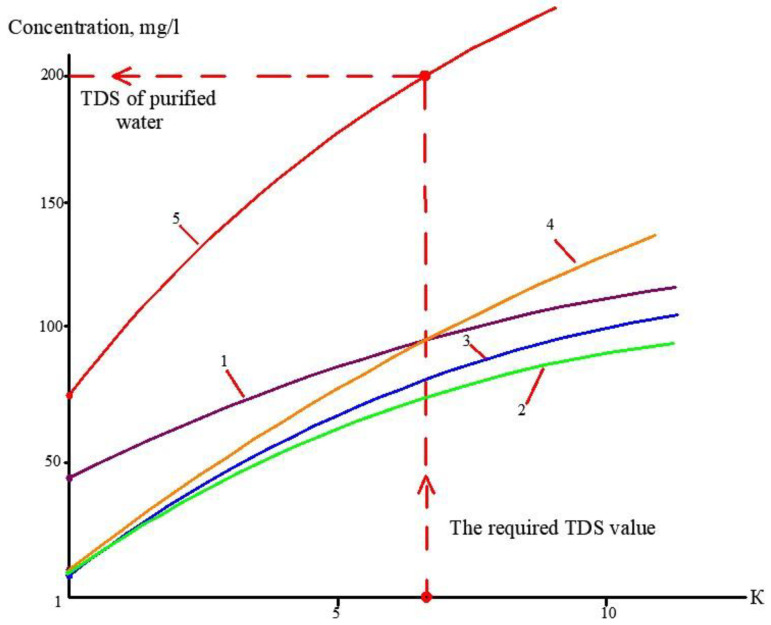
Dependencies of concentrations of different species in the reverse osmosis membrane product water on K values in the first stage of membrane treatment, and evaluation of the required recovery value: 1—chloride; 2—sodium; 3—calcium; 4—sulphate; 5—TDS.

**Figure 12 membranes-13-00153-f012:**
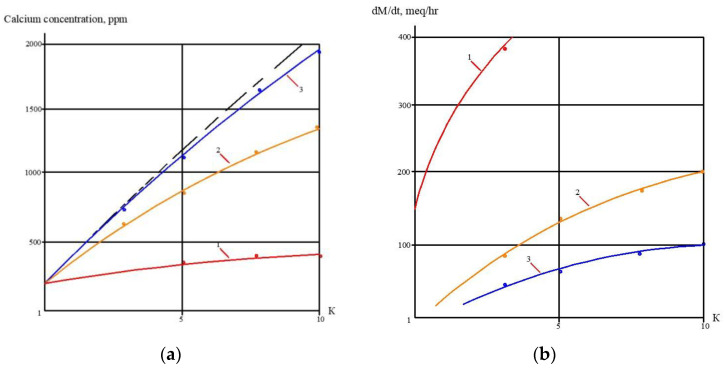
Results of calcium carbonate growth rate evaluation in the first stage of membrane treatment: the selection of membrane type for the first stage: dependencies of calcium concentration (**a**) and calcium carbonate scaling rates (**b**) on coefficient k values. 1—low pressure reverse osmosis membrane without antiscalant addition; 2—reverse osmosis membrane, antiscalant dose 5 ppm, 3—nanofiltration membrane, antiscalant dose 5 ppm.

**Figure 13 membranes-13-00153-f013:**
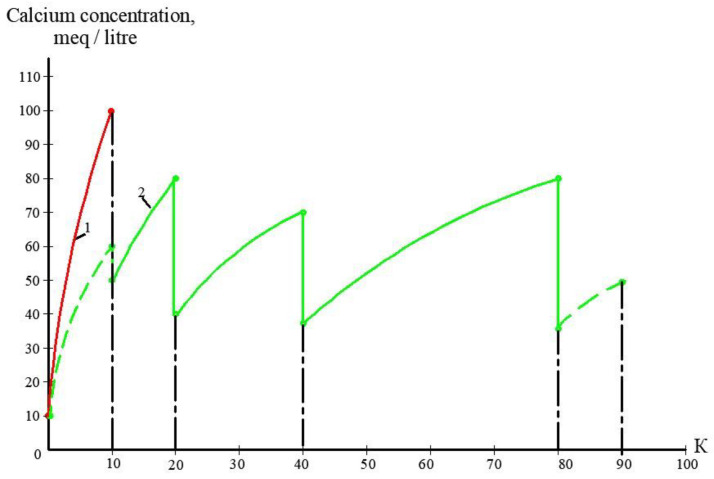
Calcium concentration change throughout the feed water concentrating cycle in the first stage of membrane treatment as dependence on K value: 1—reverse osmosis membrane module; 2—nanofiltration membrane module.

**Figure 14 membranes-13-00153-f014:**
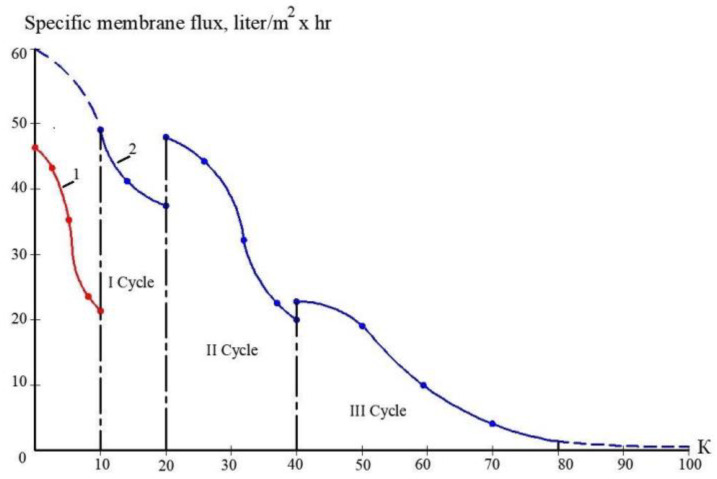
Reduction of specific membrane flux values throughout concentrating cycle as dependence on K value: 1—reverse osmosis membrane module; 2—nanofiltration membrane module.

**Figure 15 membranes-13-00153-f015:**
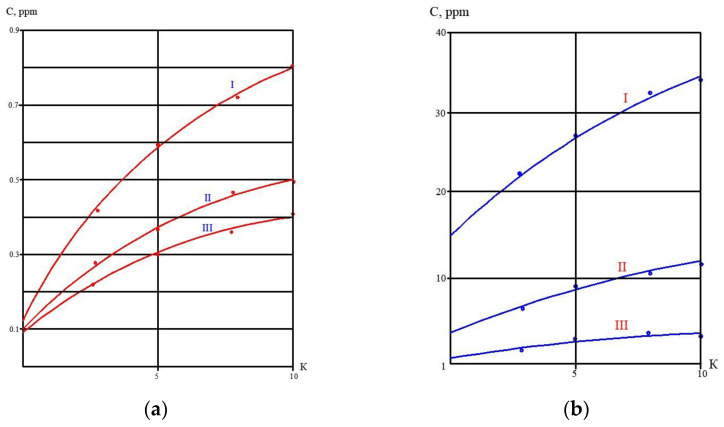
Dependencies of copper and ammonia concentration values on K in nanofiltration membrane concentrate (a) and permeate (b) after first, second and the third dilution/concentrating cycles: I—in the first cycle; II—in the second cycle after first dilution of the I cycle concentrate; III—in the third cycle after the second dilution of the II cycle concentrate.

**Figure 16 membranes-13-00153-f016:**
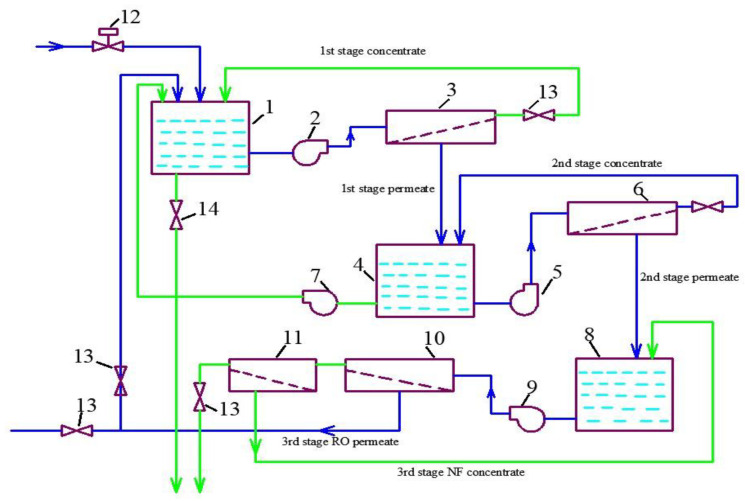
A flow diagram of the process of concentrate separation into two concentrates containing ammonia and copper: 1—membrane plant concentrate collection tank; 2—working pump for the I, II and III concentrating cycles of membrane plant concentrate; 3—NF membrane module (1st stage) for concentrating cycles; 4—NF membrane module product water collection tank; 5—working pump for the NF product water I, II and III concentration cycles; 6—NF membrane module (2nd stage) for membrane 3 product water concentrating; 7—the pump for membrane 6 concentrate supply to tank 1; 8—product water of NF membrane 6 collection tank; 9—working pump of the membrane facility for the final product water concentrating; 10—RO module (3rd stage) for product water concentrating; 11—third stage NF membrane module for recovery increase; 12—a valve for deionized water supply for concentrate diluting; 13—pressure regulation valves.

**Table 1 membranes-13-00153-t001:** Mine water composition.

Components, Units of Measurement	Concentration Mine Water	Discharge Regulation Value
Suspended matter, mg/L	48	5.25
pH	76	6.5–8.5
Calcium, Ca^2+^, ppm	278	180
Magnesium, Mg^2+^, ppm	54	40
TDS, mg/L	1520	1000
Iron, Fe, ppm	0.0023	0.1
Ammonia, ${NH}_{4}^{+}$, ppm	2.1	0.5
Sulphates, ${SO}_{4}^{2-}$, ppm	70	100
Chloride, ${Cl}^{-}$, ppm	50	-
Bicarbonate, ${HCO}_{3}^{-}$, ppm	120	-
COD, mg/L	20.6	15
Copper, ${Cu}^{2+}$, ppm	0.0073	0.001
Sodium, ${Na}^{+}$, ppm	53	-

## Data Availability

Data are available in publications.

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
