# Peer review of "Treatment of Mine Water with Reverse Osmosis and Concentrate Processing to Recover Copper and Deposit Calcium Carbonate"

_membranes, 2023, doi:10.3390/membranes13020153_

Round 1

Reviewer 1 Report

Views and comments on “Treatment of mine water with reverse osmosis and concentrate
processing to recover copper and deposit calcium carbonate”,

This paper tries to evaluate the use of RO process and concentrate processing technique to recover copper and deposit calcium carbonate for treating mine water. The paper cannot be considered for publication in its current status. However, it might be suitable after addressing the following concerns properly and precisely.

1-       Dewatering sludge for reducing operational costs and better application of the treatment system is a common and well-known system. How do the authors used this analogy for their claim regarding the contribution of this paper. Please justify this matter.

2-       The structure of the paper is misleading. The introduction section does not cover the materials needed t be in an introduction section. It can be segregated to more sections.

3-       Please clearly specify the novelty of your work.

4-       There are some serious problems about the figures. For instance, in Figure 11, what is the caption of the y-axis? I am worried that the figures have not been prepared specially for this paper and have been copied from a related report of so. Please make sure that they are original and improve the representation of them.

5-       I could see that the data availability is “Not applicable”. Since you have reported the results, why is so?

6-       Consider updated references in your work.

7-       How do the authors defend their claims for cost reduction? Are there any quantitative measures that can be presented?

8-       What was the reason for taking into account the TDS parameter for evaluating the results. What parameters can be used to address the variations of heavy metals and organic pollutants? Why they have not been reported.

9-       The paper lacks a good and constructive discussion conveying the 1) pros and cons of this treatment system, and 2) similarities and differences of this work and similar treatment sites.

10-    If possible, add some photos of the project besides the schematic pictures.

11-    There are some minor English writing problems. Please check the entire text.

Author Response

Dear reviewer,

I would like to thank you for your valuable comments which helped to improve the quality of our article. Please find a detailed description of the comments and their consideration in the article below.

Comment 1: Dewatering sludge for reducing operational costs and better application of the treatment system is a common and well-known system. How do the authors used this analogy for their claim regarding the contribution of this paper. Please justify this matter.

Reply to the comment 1: Authors agree that sludge dewatering is a common and well known system. Authors do not apply any innovations to the sludge dewatering system. The main concern of the mining water handling and management is it's treatment to reduce salts, heavy metals and organic pollutant with the aim to escape natural water pollution. This problem is efficiently solved by application of reverse osmosis method. But the main problem, a main "disadvantage" of RO is concentrate discharges. Mine water flows have very high flow rates, and amounts of concentrate are also very large. Up to present time the state of the art techniques do not suggest any efficient solution except the ZLD process.

Comment 2: The structure of the paper is misleading. The introduction section does not cover the materials needed to be in an introduction section. It can be segregated to more sections.

Reply to comment 2: Authors appreciate this comment and definitely agree, that the introduction can be extended and more details added 5 subheadings for the Introduction section.

Comment 3: Please clearly specify the novelty of your work.

Reply to comment 3: Authors thank the Reviewer for this comment. The novelty is better specified. We have added:

Concentrate disposal and possible utilization still remains unsolved problem for many cases when concentrate discharge is not available. There also cases, where existing developed techniques to reduce and eliminate concentrate is extremely expensive and the project cannot afford this approach due to high costs and availability of power and chemicals. The company that have ordered this project had only one solution based on the use of RO for mine water purification and the use of the ZLD process to dispose of concentrate discharge. Assuming the location of the project in the North, high energy costs and unavailability of reagents delivery, the customer looked for the new solution of the problem.

To solve the problem of concentrate discharge and to escape additional costs for it, softening and evaporation (unlikely the ZLD method), the authors have implemented four new techniques. This approach consists of: reducing concentrate flow by 100-200 times and hiding this volume in the dewatered sludge, thus withdrawing all rejected impurities together with dewatered sludge as the sludge moisture. To reach extremely high recoveries and escape calcium carbonate deposition in membrane channels, a new approach is developed that involves constant deposition of calcium carbonate on “seed” crystals without constant addition of softening reagents (such as lime). To efficiently utilize concentrate, another new approach is used that enables us to separate concentrate flow into two concentrated flows: a flow primarily containing monovalent ions and flow containing heavy metals.

Comment 4: There are some serious problems about the figures. For instance, in Figure 11, what is the caption of the y-axis? I am worried that the figures have not been prepared specially for this paper and have been copied from a related report of so. Please make sure that they are original and improve the representation of them.

Reply to comment 4: Authors thank for this comment. All the Figures are new and are prepared exactly for this paper, except flow diagrams of previously developed processes. We apologize for mistakes and incorrections made during preparation of our article as authors speak different languages. Figure 11 is improved according to Reviewer recommendations.

Comment 5: I could see that the data availability is “Not applicable”. Since you have reported the results, why is so?

Reply to comment 5: Authors apologize for missing this item, we just missed it. We added: "Data is available in publications".

Comment 6: Consider updated references in your work.

Reply to comment 6: Yes, we have added 6 new references.

Comment 7: How do the authors defend their claims for cost reduction? Are there any quantitative measures that can be presented?

Reply to comment 7: Authors thank the Reviewer for this valuable comment. We did not add economical discussions due to the difference in costs in our country and Europe. But economical grounds should be added to the article. We added:

Reduction of costs was the main goal of the research. According to initial (conventional) approach, the mine water was clarified using coagulants then was treated by RO to remove contaminants. RO concentrate was then softened to remove calcium and further concentrated by 10-20 times to reach the amount that does not exceed 3-5 per cent of the initial feed water flow. Then concentrate was evaporated with the end product of wet salts that were forwarded to a special landfill together with dewatered sludge after clarification stage. Such an approach involves a substantial operational costs: to buy reagents (about 300 grams of lime per one cubic meter of the feed water, about 0.3 USD per 1 cubic meter); to spend additional power (about 22-25 kilowatt per one cubic meter of evaporated concentrate) that corresponds to additional use of 1.5-2 kilowatt per 1 cubic meter of the feed water, which corresponds to additional costs of 0.25 USD. Total savings per 1 cubic meter of the feed water treatment is 0.55 USD.

The new described approach proposed by the authors use either lime softening process, nor evaporation. The described deposition of calcium on the “seed” requires less than 2 per cent of the stoichiometric amount [2].

Comment 8: What was the reason for taking into account the TDS parameter for evaluating the results. What parameters can be used to address the variations of heavy metals and organic pollutants? Why they have not been reported.

Reply to comment 8: The TDS is the main parameter to account when deciding material balance and select membranes. Membrane product flow entirely depends on TDS. Heavy metals are rejected proportionally to other salts. For our technique, the main important parameter is TDS and total amounts of salts discharged. When discussing material balance, TDS plays a major role. We have added the following:

The idea to “hide” RO concentrate in dewatered sludge and in calcium carbonate slurry is based on material balance of salts (or other impurities contained in water) that enter the dewatering facility (thickening tank) and salts withdrawn from the water treatment plant together with dewatered sludge. All dissolved salts are supplied with RO concentrate, thus, the concentrate flow rate should not exceed the flow rate of water withdrawn with the sludge as the sludge moisture. Thus, the concentrate flow rate should correspond to water flow rate withdrawn with the sludge and all dissolved salts should be withdrawn with this water amount. Figure 4 shows this balance for an example when feed water flow equals to 1000 cubic meters per hour. Usually, water clarification plant provides 5 cubic meters of dewatered sludge per 1000 cubic meters of the treated feed water. The dewatered sludge moisture value is 80 per cent that means that in dewatered sludge 4 cubic meters of water per 1 ton of solids are present. Thus, the concentrate volume should correspond to 0.4 per cent of the feed water volume. In other words, the feed water volume should be reduced (or concentrated) by 250 times by RO. For our case where feed water TDS is 1000 ppm, we can reach 120-150 grams per liter in concentrate, using pressure value of 16 Bars [1,2], or reduce feed water flow by 120-130 times. Thus, 0.4 per cent of concentrate (with 120000 ppm TDS ) is withdrawn together with the dewatered sludge and other 0.4 per cent are withdrawn together with dewatered slurry (calcium carbonate sludge).

Comment 9: The paper lacks a good and constructive discussion conveying the 1) pros and cons of this treatment system, and 2) similarities and differences of this work and similar treatment sites.

Reply to comment 9: We agree. Authors added some necessary abstracts for discussion:

Treatment of mine water remains the unsolved problem. It is partly solved by ap-plication of RO as this method provides efficient solution of heavy metals and some other pollutant removal. But treatment by RO provides a problem unsolved which is concen-trate disposal. A number of projects are considered that use the published ZLD techniques that requires evaporation of concentrate. Meanwhile, to efficiently evaporate concentrate with minimal power costs, full softening of concentrate is used (removal of calcium) which utilizes lime consumption and pellet reactors. For many customers located in far North such reagent deliveries as well as power consumption seem very expensive.

In our project the company that serves Norilsk-Nickel has addressed us to develop an alternative solution to treat RO concentrate without high expenses on reagent deliveries and evaporation. Initially the project involved ZLD process: reagent softening of RO concentrate and further evaporation to reach wet salts. To solve the problem, authors have used three techniques developed and described previously [2-4], that enable us to reduce and “hide” RO concentrate:

- reduction of concentrate volume to a value that does not exceed 0.4-0.5 per cent of the initial feed water volume and withdrawal of concentrate together with the dewatered sludge;

- reduction of concentrate volume and deposition of calcium carbonate using the “seed” crystals;

- reduction of concentrate volume using cascade  of low rejection membranes ;

- separation of concentrate into two solutions : monovalent salts and divalent salts concentrated solution to facilitate it’s further utilization.

Figures 5 (a and b) show the steps to reach high TDS of concentrate and to reduce it’s volume. Figures 10 and 11 shows values of different ions and impurities concentration values that enable us to predict chemical composition. Calcium carbonate deposition on the “seed” crystals reduces TDS and facilitates further concentration increase. Experiments show that high TDS can be reached and concentrate volume can be reduced by a value of less than 1 per cent of initial volume. This amount can be shared between sludge dewatering facilities for suspended sludge dewatering and calcium carbonate precipitate. Figure 4 shows flow diagrams and mass (salt) balance during sludge dewatering to provide withdrawal of all excessive salts and rejected impurities together with dewatered aluminum sludge and precipitated calcium carbonate.

The proposed approach provides treatment of mine water with reverse osmosis plant and utilization of concentrate by reduction of it’s volume to the value less than 0.5-1 per cent of initial feedwater volume and withdrawal of rejected salts and impurities with dewatered sludge and slurry.

Economical considerations of the project accounts for savings provided by new technology of concentrate utilization. The approach does not provide liquid wastes as separates mine water into clean purified water flow discharged onto water body and dewatered sludge that is forwarded to landfill. The technological scheme lacks chemical softening and evaporation of concentrate. This approach saves operational costs.

Separation of concentrate and separation of aluminum sludge and slurry permits to use the sludges separately to save payments for special landfills with category.

Reduction of costs was the main goal of the research. According to initial (conven-tional) approach, the mine water was clarified using coagulants then was treated by RO to remove contaminants. RO concentrate was then softened to remove calcium and further concentrated by 10-20 times to reach the amount that does not exceed 3-5 per cent of the initial feed water flow. Then concentrate was evaporated with the end product of wet salts that were forwarded to a special landfill together with dewatered sludge after clarification stage. Such an approach involves a substantial operational costs: to buy reagents (about 300 grams of lime per one cubic meter of the feed water, about 0.3 USD per 1 cubic meter); to spend additional power (about 22-25 kilowatt per one cubic meter of evaporated concentrate) that corresponds to additional use of 1.5-2 kilowatt per 1 cubic meter of the feed water, which corresponds to additional costs of 0.25 USD. Total savings per 1 cubic meter of the feed water treatment is 0.55 USD.

The new described approach proposed by the authors use either lime softening process, nor evaporation. The described deposition of calcium on the “seed” requires less than 2 per cent of the stoichiometric amount [2].

Comment 10: If possible, add some photos of the project besides the schematic pictures.

Reply to comment 10: We agree, and added photos of the test units. Figures 1, 2 and Figure 6.

Comment 11: There are some minor English writing problems. Please check the entire text.

Reply to comment 11: Authors apologize for some text incorrections and worked on its improvement.

Reviewer 2 Report

The current manuscript describes the “Treatment of mine water with reverse osmosis and concentrate processing to recover copper and deposit calcium carbonate”. The present study demonstrated the technique involves concentrate reduction and withdrawal with dewatered sludge. The different parameters have been checked and performance has been noted. After checking the whole manuscript the reviewers found that there are a lot of scopes to improve the manuscript. However, some of the explanations may need further illustration; and more importantly, because this type of work has been already reported by a lot of researchers using different types of nanomaterials. One important point is that the authors of the abstract have to be revised. Also, the authors should consider critically these comments to improve the quality of the work. 

Few examples of where authors should make changes are provided below:

  1. The abstract is not showing the exact output and scope. Need to rewrite the abstract.
  2. The introduction particularly 2nd paragraph can be improved by adding recent literature on heavy metals removal and will get an idea about the mechanism like Chem. Eng. J. 446 (2022) 137303; J. Membr. Sci., 609 (2020) 118212 etc, and revised the incomplete sentences.
  3. The figure resolution should be increased. The text should be more clear.
  4. All equations need to recheck once.
  5. Avoid the sentence starting with the word “And”.
  6. How to maintain a material balance need to explain in detail.
  7. All material details should be discussed in the manuscript.
  8. Whenever required the x and y-axis legends should be there on the figures.
  9. Why only consider copper not other heavy metals?
  10. Polish the grammar once.

Author Response

Dear reviewer,

I would like to thank you for your valuable comments which helped to improve the quality of our article. Please find a detailed description of the comments and their consideration in the article below.

The current manuscript describes the “Treatment of mine water with reverse osmosis and concentrate processing to recover copper and deposit calcium carbonate”. The present study demonstrated the technique involves concentrate reduction and withdrawal with dewatered sludge. The different parameters have been checked and performance has been noted. After checking the whole manuscript the reviewers found that there are a lot of scopes to improve the manuscript. However, some of the explanations may need further illustration; and more importantly, because this type of work has been already reported by a lot of researchers using different types of nanomaterials. One important point is that the authors of the abstract have to be revised. Also, the authors should consider critically these comments to improve the quality of the work.

Few examples of where authors should make changes are provided below:

Comment 1: The abstract is not showing the exact output and scope. Need to rewrite the abstract.

Reply to the comment 1: Authors agree and the abstract is revised.

Comment 2: The introduction particularly 2nd paragraph can be improved by adding recent literature on heavy metals removal and will get an idea about the mechanism like Chem. Eng. J. 446 (2022) 137303; J. Membr. Sci., 609 (2020) 118212 etc, and revised the incomplete sentences.

Reply to the comment 2: Authors thank the Reviewer and added the new references:

  1. Sachin Karki, Pravin G.Ingole. Development of polymer-based new high performance thin film nanocomposite nanofiltration membranes by vapor phase interfacial polymerization for the removal of heavy metal ions. Chemical Engineering Journal, Volume 446, Part 3, 2022, 137303.
  2. Leon Fuks, Agnieszka Miskiewicz and Grazyna Zakrzewska-Koltuniewicz. Sorption-Assisted Hybrid Method for Treatment of the redioactive Aqueous Solutions. Chemistry 2022, 4(3), 1076-1091; https://doi.org/10.3390/chemistry4030073
  3. Moucham Borpatra Gohain, Radheshyam R.Rawar. Development of thin film nanocomposite membrane incorporated with mesoporous synthetic hectorite and MSH@UiO-66-NH2 nanoparticles for efficient targeted feeds separation, and antibacterial performance. Journal of Membrane Science 609:118212. DOI:10.1016/j.memsci.2020.118212
  4. Zahra Samavati, Alireza Samavati, Pei Sean Goh, Ahmad Fauzi Ismmail, Mohd Sohaimi Abdullah. A comprehensive review of recent advances in nanofiltration membranes for heavy metal removal from wastewater. Chemical Engineering Research and Design, Volume 189, January 2023, pp. 530-571.
  5. Kailash Khube, T. Matsuura. Removal of heavy metals and pollutants by membrane adsorption techniques. Applied Water Science, 2018, 8(1); DOI:10.1007/S13201-018-0661-6.
  6. Thi Sinh Vo, Muhammad Mohsin Hossain, Hyung Mo Jeong, Kyunghoon Kim. Heavy metal removal applications using adsorptive membranes. Nano Convergence, 7, 36(2020).

Comment 3: The figure resolution should be increased. The text should be more clear.

Reply to the comment 3: Authors thank the Reviewer for this comment and have done their best to improve the text a d drawings.

Comment 4: All equations need to recheck once.

Reply to the comment 4: All equations are rechecked.

Comment 5: Avoid the sentence starting with the word “And”.

Reply to the comment 5: Authors agree and feel ashamed for such a mistake.

Comment 6: How to maintain a material balance need to explain in detail.

Reply to the comment 6: Material balance flow diagram is shown on Figure 4. We added the following explanation:

The idea to “hide” RO concentrate in dewatered sludge and in calcium carbonate slurry is based on material balance of salts (or other impurities contained in water) that enter the dewatering facility (thickening tank) and salts withdrawn from the water treatment plant together with dewatered sludge. All dissolved salts are supplied with RO concentrate, thus, the concentrate flow rate should not exceed the flow rate of water withdrawn with the sludge as the sludge moisture. Thus, the concentrate flow rate should correspond to water flow rate withdrawn with the sludge and all dissolved salts should be withdrawn with this water amount. Figure 4 shows this balance for an example when feed water flow equals to 1000 cubic meters per hour. Usually, water clarification plant provides 5 cubic meters of dewatered sludge per 1000 cubic meters of the treated feed water. The dewatered sludge moisture value is 80 per cent that means that in dewatered sludge 4 cubic meters of water per 1 ton of solids are present. Thus, the concentrate volume should correspond to 0.4 per cent of the feed water volume. In other words, the feed water volume should be reduced (or concentrated) by 250 times by RO. For our case where feed water TDS is 1000 ppm, we can reach 120-150 grams per liter in concentrate, using pressure value of 16 Bars [1,2], or reduce feed water flow by 120-130 times. Thus, 0.4 per cent of concentrate (with 120000 ppm TDS ) is withdrawn together with the dewatered sludge and other 0.4 per cent are withdrawn together with dewatered slurry (calcium carbonate sludge).

Comment 7: All material details should be discussed in the manuscript.

Reply to the comment 7: We appreciate the Reviewer for this comment and added the following:

Reduction of costs was the main goal of the research. According to initial (conven-tional) approach, the mine water was clarified using coagulants then was treated by RO to remove contaminants. RO concentrate was then softened to remove calcium and further concentrated by 10-20 times to reach the amount that does not exceed 3-5 per cent of the initial feed water flow. Then concentrate was evaporated with the end product of wet salts that were forwarded to a special landfill together with dewatered sludge after clarification stage. Such an approach involves a substantial operational costs: to buy reagents (about 300 grams of lime per one cubic meter of the feed water, about 0.3 USD per 1 cubic meter); to spend additional power (about 22-25 kilowatt per one cubic meter of evaporated concentrate) that corresponds to additional use of 1.5-2 kilowatt per 1 cubic meter of the feed water, which corresponds to additional costs of 0.25 USD. Total savings per 1 cubic meter of the feed water treatment is 0.55 USD.

The new described approach proposed by the authors use either lime softening process, nor evaporation. The described deposition of calcium on the “seed” requires less than 2 per cent of the stoichiometric amount [2].

Comment 8: Whenever required the x and y-axis legends should be there on the figures.

Reply to Comment 8: We apologize and made necessary corrections.

Comment 9: Why only consider copper not other heavy metals?

Reply to Comment 9: Copper was the only heavy metal, causing concern to the customer, as other heavy metals have low content. Authors refrained from other expensive analysis of other metals and concentrated on copper discussion.

Comment 10: Polish the grammar once.

Reply to Comment 10: We thank the Reviewer and tried to improve our grammar.

Round 2

Reviewer 1 Report

Dear Editor of Membranes Journal,

Authors have appropriately addressed my concerns.

The paper is suitable for publication in its present form.

Best regards.

Reviewer 2 Report

In my opinion authors have made the required changes in the manuscript so it can be accepted for the publication.